# Optimal Sand−Paving Parameters Determination of an Innovatively Developed Automatic Maize Seeding Machine

Bohan Fu [1,2,3], Weizhong Sun [1,2,3] and Zhao Zhang [1,2,3,4,*]

1    Key Laboratory of Smart Agriculture System Integration, Ministry of Education, Beijing 100083, China;
     2020308130329@cau.edu.cn (B.F.); 2020308130208@cau.edu.cn (W.S.)
2    Key Laboratory of Agricultural Information Acquisition Technology, Ministry of Agriculture and Rural
     Affairs, China Agricultural University, Beijing 100083, China
3    College of Information and Electrical Engineering, China Agricultural University, Beijing 100083, China
4    Department of Agricultural and Biosystems Engineering, North Dakota State University,
     Fargo, ND 58102, USA
*    Correspondence: zhaozhangcau@cau.edu.cn

**Abstract:** Maize is an important crop to ensure food safety. High-quality seeds can guarantee a good yield. The maize seed germination rate is the most important information for the maize industry, which can be obtained through the seed germination test. An essential stage in determining the germination rate is the planting of the seeds. The current seed planting process is fully manual, which is labor-intensive and costly, and it requires the development of an autonomous seeding machine. This research developed an automatic maize seeding machine, consisting of four operations: paving sand, seed layout, watering, and covering the seed. Among the four procedures, sand paving is a crucial step, the performance of which is affected by the gate opening size, conveyor speed, and sensor mounting location. Three performance evaluating factors are the weight of sand in the tray, the volume of sand left on the conveyor, and sand surface flatness. A full factorial experiment was designed with three variables and three levels to determine an appropriate factor combination. RGB-D information was used to calculate the volume of sand left on the conveyor and sand flatness. An analytic hierarchy process was employed to assign weights to the three evaluation indicators and score the various combinations of factors. The machine for paving sand achieved a satisfactory result with an opening size of 10.8 mm, a sensor distance of 9 cm, and a conveyor belt speed of 5.1 cm/s. With the most satisfactory factors determined, the machine shows superior performance to better meet practical applications.

**Keywords:** automatic maize seeding machine; sand-paving device; image processing; analytic hierarchy process

## 1. Introduction

Maize production plays a crucial role in ensuring global food security [1], and China holds the position of the second-largest maize producer worldwide [2]. The germination rate of seeds holds significant importance as it directly impacts the overall yield, serving as a vital indicator of seed quality. In China's seed production industry, Standard GB/T 5243.4 is employed to determine the seed germination rate, encompassing both seed planting and seedling counting processes. While computer vision has been utilized in various studies for automated seedling counting [3,4], seed planting remains a fully manual process. The manual sowing procedure involves several distinct steps. The process starts with the operator preparing the tray. Firstly, they cover the tray with sand and then flatten it to create a smooth surface (Step 1). Next, the operator evenly distributes the seeds onto the sandy surface of the tray. To aid germination, they spray water onto the seeds (Step 2). In order to ensure optimal conditions for seed growth, another layer of sand is then added on top of the seeds, providing additional protection and support (Step 3). Finally, the

prepared tray is placed into an incubator that creates a suitable environment to promote seed germination (Step 4). The subjective nature of operators can have an impact on the results of the experiment [5–7]. Moreover, the repetitive and extended tasks of sand flattening and sowing can pose occupational health risks for workers, including back and wrist injuries [8]. Therefore, there is a need for the development of an autonomous maize seeding machine.

This study designed and developed a maize seed sowing machine to automatize the seeding process, which encompasses four modules: sand-paving tray, seed release mechanism, water spraying, and covering the seed with sand. The initial step involves paving the tray with sand, followed by the crucial task of leveling or flattening the sand. Without proper flattening, when seeds are released onto the sand, there is a risk of them rolling away from their intended position, disrupting the predetermined uniform pattern. This disruption can lead to challenges in accurately counting and assessing the stand, as multiple seeds may germinate in close proximity. Therefore, maintaining a flat sand surface is essential, as it ensures the desired uniformity for successful seed distribution and germination.

The levelness of the sand surface directly influences seed germination. In a study by researchers [9], different soil-covering and compacting devices were compared, and factors such as soil morphology and compaction were evaluated. Xu et al. (2021) developed a mathematical model to describe the relationship between soil output, mulch disc depth, and mulch disc helical blades [10]. To overcome the subjectivity and inaccuracy associated with manual observation, this experiment employs an RGB-D camera to quantitatively assess sand volume and flatness [11]. It has been demonstrated by Zhou et al. (2021) that RGB-D cameras can effectively capture RGB and depth data [12]. These cameras have gained wide adoption in agriculture and have shown favorable results in various practical applications, including plant phenotyping [13], 3D segmentation of plants, and obstacle detection for agricultural robots [14,15]. Therefore, utilizing an RGB-D camera for evaluating the sand surface represents a promising option.

With the ultimate objective of achieving practical implementation of the automatic maize seeding machine, this study aimed to explore the impact of three factors, namely gate opening size, sensor mounting location, and conveyor speed, on the evaluation of sand surface flatness. The goal was to determine the optimal combination of parameters for effective practical application.

## 2. Materials and Methods

### 2.1. Brief Introduction to the Automatic Maize Seeding Machine

The automatic maize seeding machine, shown in Figure 1, has been innovatively designed. It comprises a conveyor belt and four distinct modules operating independently: (1) sand paving, (2) seed placement, (3) watering, and (4) sand covering. At the beginning of the process, an empty tray is positioned on one end of the machine (the left-hand side in Figure 1). As the conveyor belt transports the tray forward, its position is constantly monitored by sensors (GP2Y0A21, manufactured by Sharp Corporation in Osaka, Japan). When the tray is detected nearing the sand-paving device, the motor is activated, opening the gate to commence the sand-paving process. Upon detection of the tray leaving the sand-paving device, the motor is triggered to close the gate, signaling the end of the sand-paving process. Following the sand-paving phase, the tray proceeds to the seed placement mechanism, where seeds are released onto the sand surface. Subsequently, the tray moves forward into the watering section. Finally, the tray reaches the second sand-filling section, where the seeds are covered with sand.

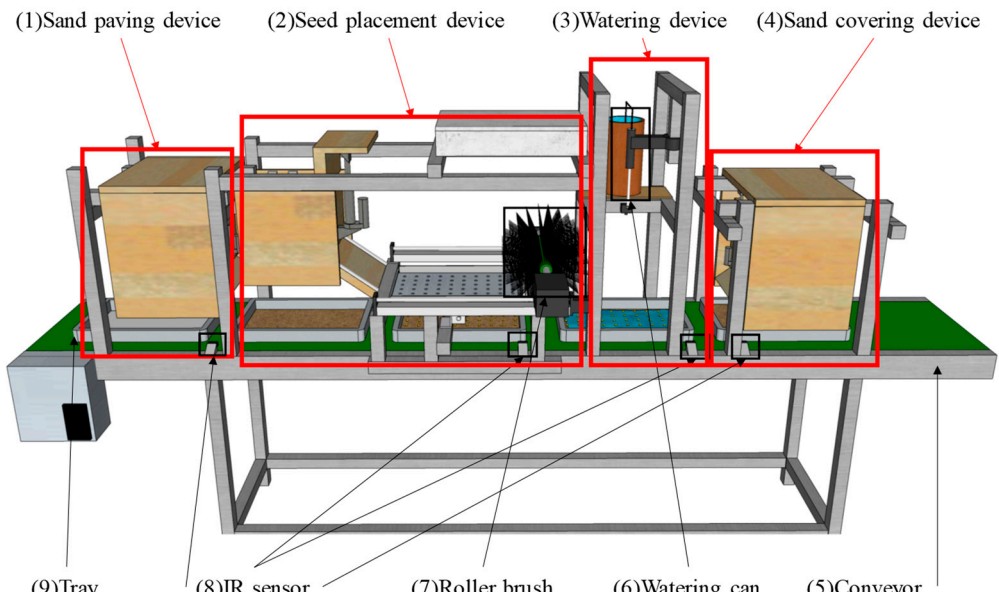

**Figure 1.** Automatic maize seeding apparatus: (1) Sand-paving device; (2) Seed placement device; (3) Watering device; (4) Sand-covering device; (5) Conveyor for transporting trays; (6) Watering can; (7) Roller brush for sweeping the seeds; (8) IR sensor for monitoring tray location; and (9) Empty tray for seed germination test.

The automatic maize seeding machine was developed by our team to fully replace the labor for seeding. Among the four key procedures (i.e., sand paving, seed placement, watering, and sand covering), the first procedure of sand paving is a crucial step, which functions to pave the tray with sand. The evenness of the sand surface is crucial for successful sand paving. Uneven surfaces can cause seed displacement and result in seeds rolling away from their intended landing location. This can lead to inconsistencies in the final location of the seeds and negatively impact the overall quality of the paving. The sand-paving device plays a crucial role in determining the machine's performance. Prior to the seeds being released onto the sand, they are arranged in a desirable uniform pattern. Any disruption to this pattern can result in two seeds germinating at the same location, presenting a challenge when it comes to accurately counting the seedlings. Therefore, the efficacy of the sand-paving device is critical for ensuring successful seed distribution and subsequent seedling monitoring.

*2.2. Sand-Paving Device*

The sand-paving device (Figure 2) has a dimension of 297 mm × 290 mm × 400 mm, and the bottom plate has a slope of 45°, which would automatically move the sand downward to the tray. The gate open and close is controlled by a 24V DC motor with a max 180 mm/s linear speed and a 100 mm stroke. The control system of the paving device utilizes an Arduino board (UNO R3, Arduino, Strambino, Italy). The system operates as follows: the sensor detects the presence of the tray by measuring the distance between the sensor and the tray. When the tray enters the sensor's view, the measured distance decreases from over 10 cm to less than 10 cm. This prompts the Arduino to send a signal to the relays (JQC-3FF-S-Z 5V D.C.; Shenzhen Weixin, Shenzhen, China) in order to open the gate. Conversely, when the tray moves out of the sensor's view, the measured distance increases from less than 10 cm to over 10 cm, signaling the Arduino to close the gate by triggering the relays.

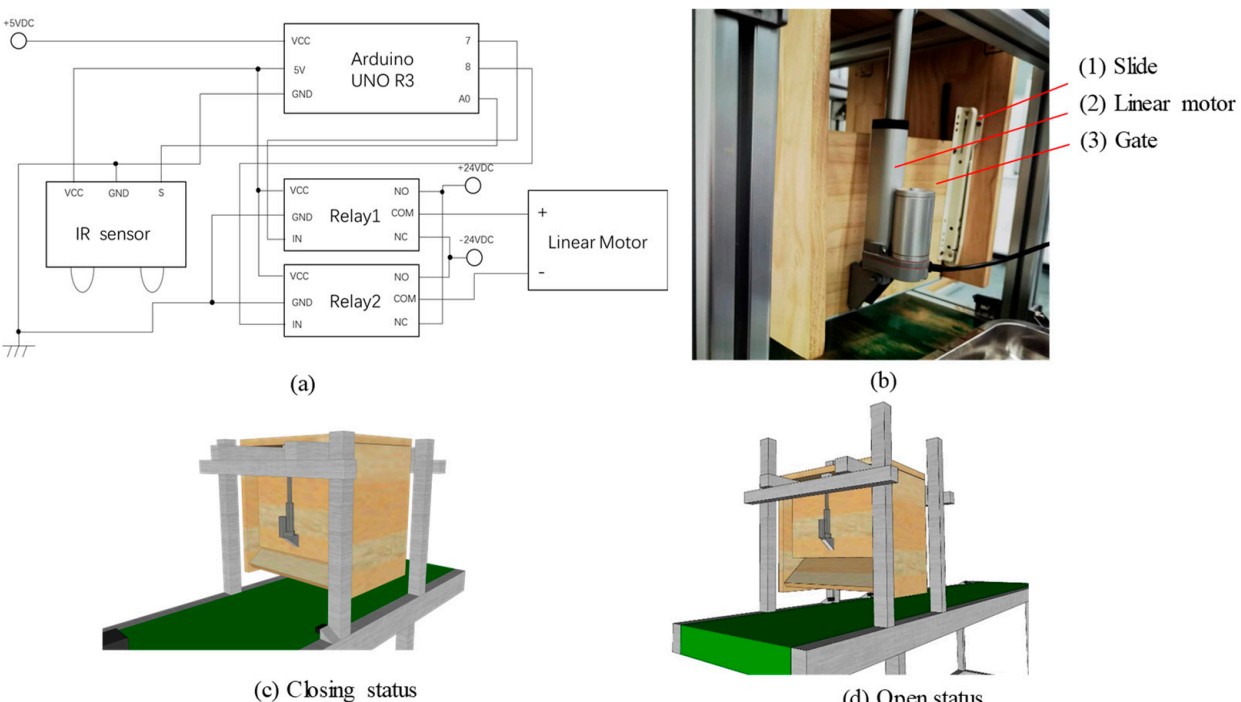

**Figure 2.** Newly developed sand−paving device: (**a**) Circuit diagram; (**b**) Sand-paving device; (**c**) gate in closed mode; and (**d**) gate in open mode.

*2.3. Experimental Procedure*

A full factorial experiment design was employed to evaluate the significance of factors and explore all possible combinations of factor levels. This approach enables a comprehensive comparison of the impact of different factor combinations on machine performance. The first factor examined is the conveyor speed, which influences the duration of the tray's movement beneath the sand gate and consequently affects the flatness of the sand surface. The second factor is the gate opening size, which directly determines the rate at which sand is filled. Lastly, the relative distance of the sensor to the sand-filling gate is examined as the third factor.

The three factors were configured with three levels each: (1) conveyor speeds of 0.13, 0.51, and 0.65 m/s; (2) gate opening sizes of 7.2, 9.0, and 10.8 mm; and (3) sensor positions relative to the sand control gate of 40.0, 70.0, and 90.0 mm (the relative distances were determined through preliminary experiments). In total, there were 27 unique settings, with each setting replicated three times, resulting in a total of 81 runs.

After each run, three parameters were recorded: the weight of the tray, the volume of sand remaining on the conveyor belt, and the flatness of the sand surface. Additionally, RGB-D information was collected using an Intel RealSense D435i camera from Santa Clara, California, U.S. These data were collected for both the tray and the sand on the conveyor belt. To obtain the tray weight, the tray was carefully moved away from the conveyor and weighed using a scale (SNJ-50001; SINUOJIE, Shenzhen, China). The image data were processed using MATLAB R2022a (The Mathworks, Inc., Natick, MA, USA). Origin 2022 (OriginLab, Inc., Northampton, Mass., USA) was used for image plotting, while third-party libraries such as NumPy and OpenCV were used in Python. Figure 3 shows the overall procedure for determining the satisfactory combination of factors.

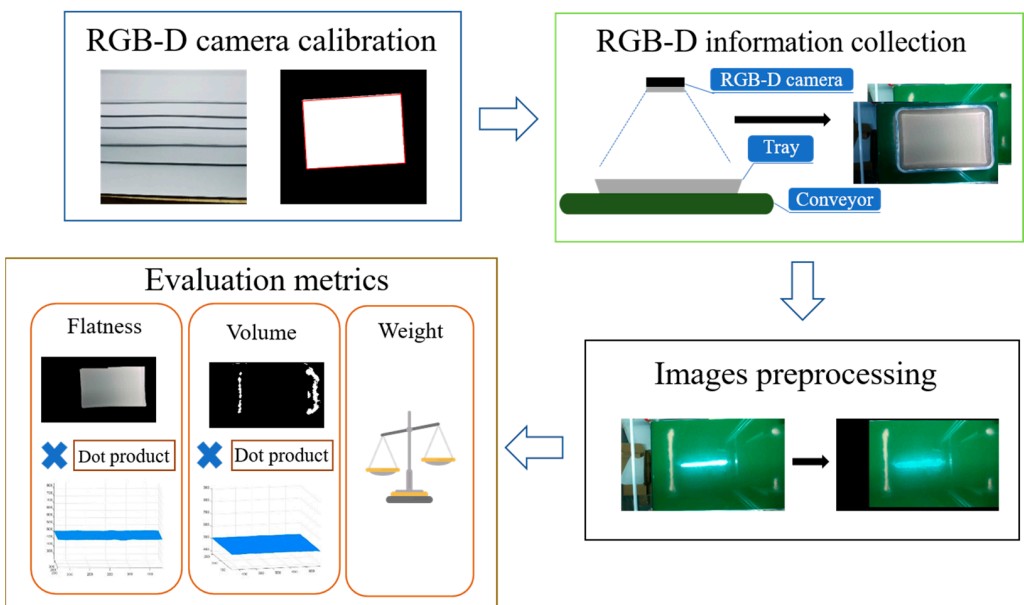

**Figure 3.** The overall procedure of determining the optimal sand-paving parameters of a newly developed automatic maize seeding machine.

*2.4. Calibration of the RGB-D Data*

2.4.1. Depth Information

The evenness of the sand surface is one of the more important topics in this research. However, directly measuring the sand surface presents challenges. In this study, RGB-D information is leveraged to measure the flatness. Before utilizing the depth information to evaluate sand evenness, a calibration process is conducted to ensure accurate sensor data.

A manually built staircase using an A4 sheet with known dimensions was used to calibrate the depth data (Figure 4). In this study, the effect of ambient light is not received by the calibration effect, as these data were collected in a dark environment. Twenty A4 sheets were used to construct each step of the stairs, and their thickness was maintained at 3 mm after measurement with a caliper. It was found that the measuring error was within 0.1 mm. Figure 4 visually demonstrates the evident staircase pattern, with a noticeable 3 mm difference in height between steps. This suggests that the depth data resolution can achieve a level of 1 mm, making it suitable for assessing the evenness of the sand surface.

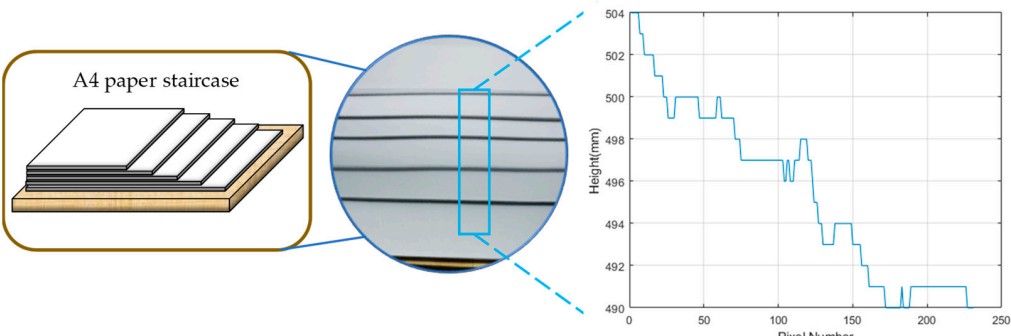

**Figure 4.** RGB-D camera calibration for depth information using A4 paper staircase.

2.4.2. RGB Image Calibration

In addition to calibrating depth information, it is necessary to check the distortion of the collected RGB images [16]. An A4 sheet was placed on the conveyor belt, and an RGB image was collected (Figure 5). A MATLAB® tool, color thresholder, was used to separate the A4 paper and extract the mask image.

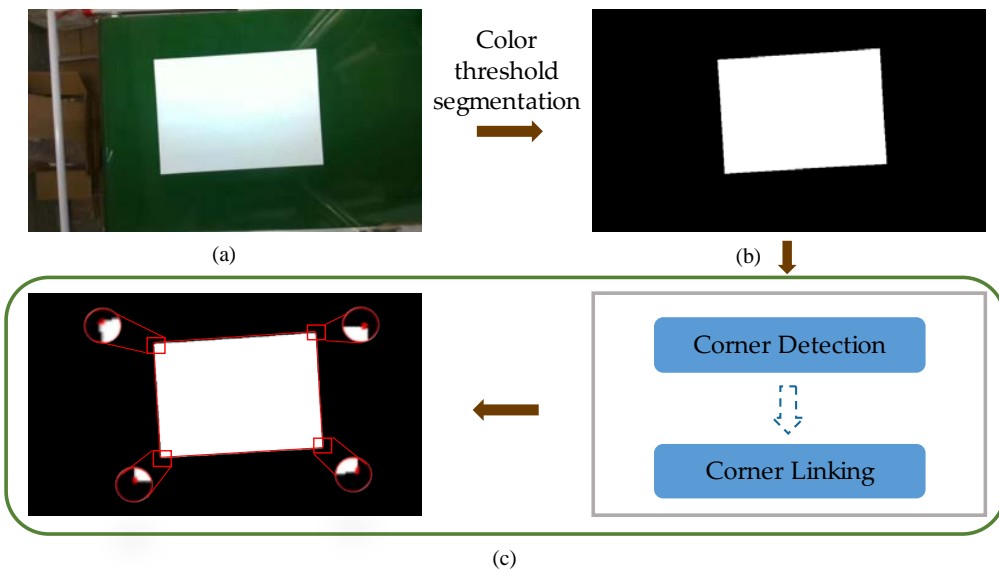

**Figure 5.** RGB image and segmentation: (**a**) Original image; (**b**) corresponding binary image after thresholding; and (**c**) four vertices of the binary image connected.

The corner Harris function, implemented in the OpenCV library, was utilized to detect the coordinates of the four vertices in the binary image. Upon computation, it was determined that the length of the upper and lower edges of the A4 paper matched that of the left and right edges. Additionally, in Figure 5c, the lines connecting the four vertices closely aligned with the boundaries of the A4 paper in the binary image. From these observations, it can be inferred that the camera-captured images exhibited minimal distortion. The actual area corresponding to a pixel can be calculated by counting the number of white pixels in image N (Figure 5c). The area corresponding to pixels was utilized in the following study as the actual volume of sand needed to be measured. The size of an A4 paper is known to be 210 mm × 297 mm. The value of 0.19 mm$^2$, obtained after calculation, represents the area corresponding to one pixel as determined by Equation (1).

$$s = \frac{N}{210 \times 297} \tag{1}$$

*2.5. Data Acquisition for the Experiment*

The RGB-D camera was installed 0.51 m above the conveyor, and the illumination was provided using two LED lights (RS-40; Shenzhen Ruisekeji, Shenzhen, China). Before each experimental run, the weight $m_1$ of the empty tray was measured, and after the sand was filled in the tray, it was transported by the conveyor belt to the view of the RGB-D camera for data acquisition. The tray was then weighed again with the sand to obtain the weight $m_2$. Thus, the weight of sand is $m_2 - m_1$. All captured RGB images and depth information tables are standardized with a size of 480 × 848 pixels, signifying that each RGB image pixel corresponds to a specific depth value. Lastly, for each trial, two RGB photos were taken: one of the trays and one of the conveyor belts (capturing the sand left on the tray), along with their corresponding depth information.

*2.6. Images Preprocessing*

In the collected RGB image (Figure 6a), there is significant noise present in the non-conveyor areas. To facilitate subsequent data processing, the noisy regions of the image were converted into black areas. In the original image, reflections caused by the conveyor belt manifest as bright areas in the picture. These white highlights pose a major challenge for target segmentation, as it becomes difficult to distinguish between the sand and the reflection in the S channel of the HSV (hue saturation value) color space (Figure 6b). To

mitigate this interference, an approach was implemented to identify the highlighted areas and replace them with the color values of the adjacent pixels, effectively eliminating the highlights in the target region of the image. Figure 6d illustrates the noticeable difference after the elimination of highlights, where a clear distinction can be observed between reflection areas and the sand in the S channel. The fast marching method image correction approach, as developed by Telea (2004) [17], was employed for this purpose. As seen in Figure 6c, this repair method demonstrates the potential for producing superior results compared to previous repair algorithms.

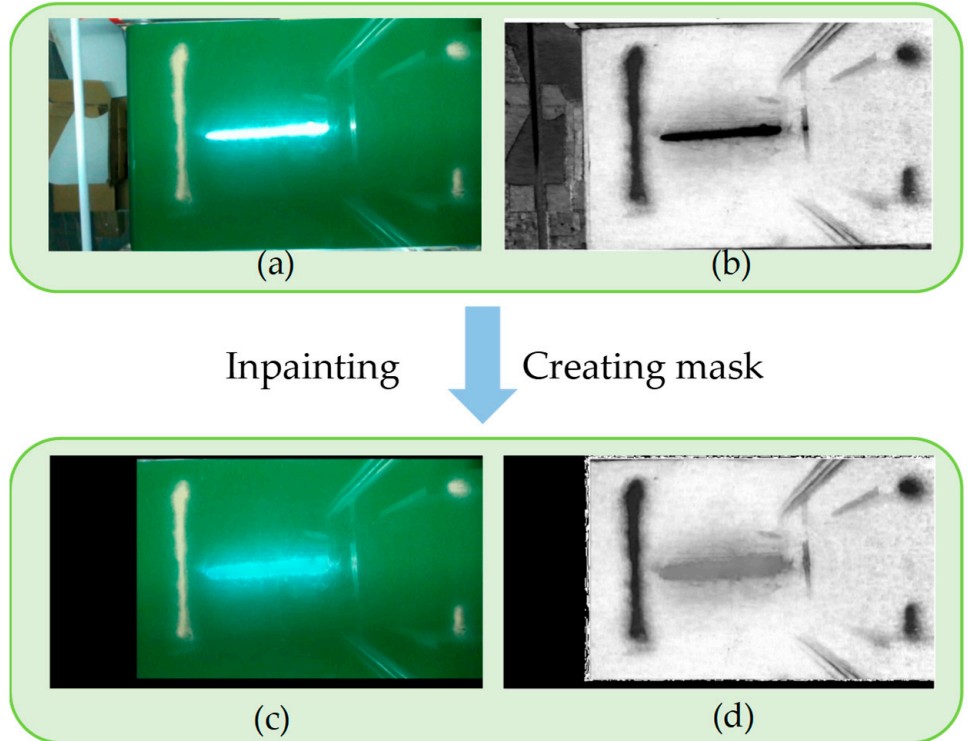

**Figure 6.** Image preprocessing: (**a**) Original image; (**b**) Original image in HSV color space; (**c**) Image after highlight repair; and (**d**) Image in HSV color space after highlight repair.

### 2.7. Evaluation Metrics

In this study, the weight of the sand in the tray, the volume of sand on the conveyor (outside the tray), and the evenness of the sand surface are the three parameters employed to determine the optimal combination of factors.

### 2.7.1. The Weight of Sand in the Tray

The weight of sand plays a crucial role in the sand-filling process and has a significant impact on seed germination [18]. It is directly measured in the experiment using an electronic scale. The weight of sand applied in each experiment is determined by weighing the empty tray and subsequently re-weighing the tray after the sand-paving process is completed. The difference between the two measurements represents the weight of sand in the tray. A higher sand weight generally ensures a favorable environment for seed germination and is therefore preferred.

### 2.7.2. The Volume of Sand Leaked onto the Conveyor Belt

The amount of sand that spills onto the conveyor belt can vary depending on the timing of the gate opening and the speed at which the conveyor moves during the sand-paving process. Any sand that falls outside the tray is considered a loss and cannot be reused. An increase in the volume of sand outside the tray leads to a proportional increase in the magnitude of loss. Therefore, when dealing with different levels of the selected

factors, measures should be taken to minimize these losses, particularly by reducing the volume of sand that falls outside the tray. The sand on the conveyor belt needs to be separated from the image, and then the heights of each pixel corresponding to the sand should be summed to calculate the amount of sand that has spilled outside the tray.

Color thresholding was applied after removing the highlights to isolate specific colors in the image (Figure 7). In this study, the HSV format was used for color thresholding. The HSV color space is more aligned with human visual perception than the RGB color space because it effectively separates intensity from hue (H) [19]. However, due to slight variations in lighting conditions within the experimental environment, there were disparities in color between the images, leading to suboptimal results in color thresholding as depicted in Figure 7(b2). Upon careful examination of sand characteristics, it was discovered that the sand could still be successfully distinguished and separated from the background.

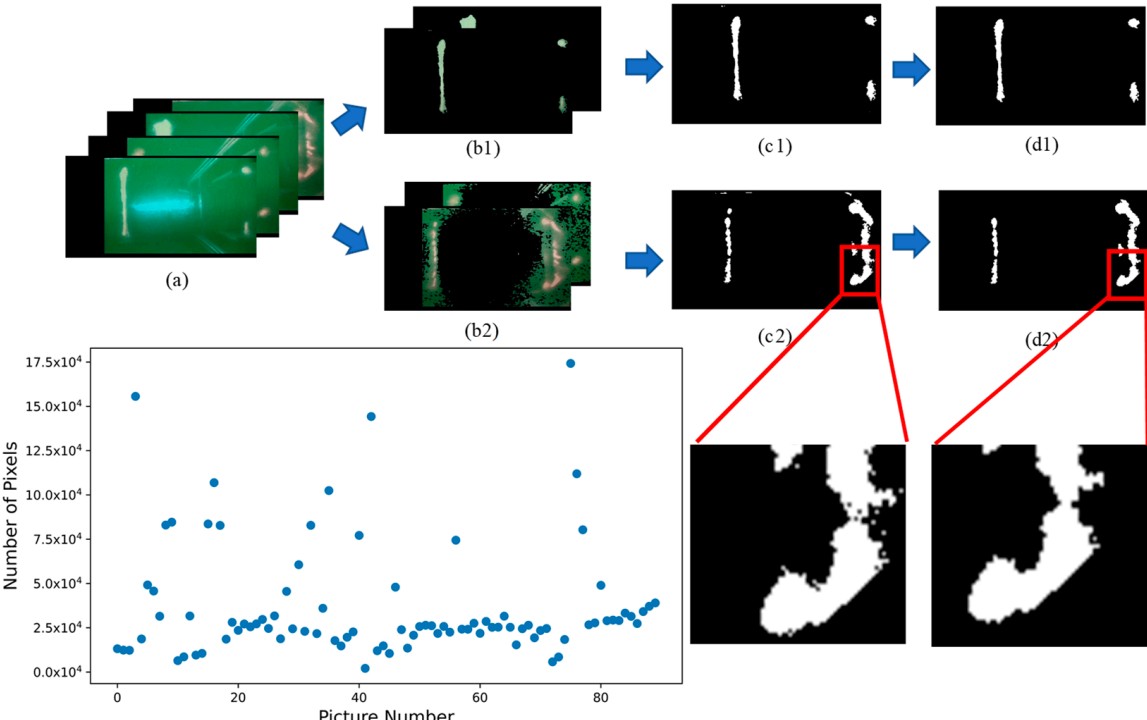

**Figure 7.** Sand area segmentation from the conveyor: (**a**) Image after preliminary preprocessing; (**b1,b2**) Results after the first color thresholding; (**c1,c2**) Binary images are generated after color thresholding with different segment methods; and (**d1,d2**) Images after noise reduction.

In contrast, the poorly separated areas occupy a relatively large area, as shown in Figure 7(b2). Therefore, the pixel values of the non-masked areas in all segmented images were counted. The area of all non-masked regions in the pictures was binary classified using K-means [20] to discriminate between the two segmentation situations (Figure 7(b1,b2)). The remaining 14 images were labeled low quantity and segmented using the YCbCr color space to produce binary images (Figure 7(c2)). A total of 76 images were directly converted into binary images (Figure 7(c1)), while the remaining 14 images were segmented directly into binary images (Figure 7(c2)). The reason for selecting this color space for segmentation is that the images clearly show uneven illumination, and the YCbCr color space is more suitable for image segmentation based on chromaticity and intensity [21]. Two sets of binary images with improved segmentation results were obtained after each set of images was processed separately, and the parameters were adjusted through continuous trial and error (Figure 7(c1,c2)). The following steps were taken to remove noise from the segmented binary images: (1) filling the 'holes' in the white areas using the built-in function in MATLAB® to fill the binary image and (2) setting all regions in the image with fewer than 500 pixels to black. Finally, the area of sand that leaked outside the tray was obtained

(Figure 7(d1,d2)) after these steps were taken. The area of each pixel in the processed images was determined during the preprocessing steps described in previous sections. Counting the number $N$ of images with white pixels (with a value of 1) is essential to determine the area $s$ of sand that has leaked outside the tray and passed (Equation (2)).

$$S = 0.19 \cdot N \tag{2}$$

To generate the depth information table (referred to as BWD) with a mask, the binary image ($BW$) representing the sand that spilled outside the tray after processing needs to undergo a dot product with the depth information table ($D$). Additionally, it is crucial to calculate the distance between the depth camera and the conveyor belt. Due to slight vibrations in the movement of the conveyor belt and the equipment during each experiment, there may be minor variations in the distance between the depth camera and the conveyor belt. In this process, the average depth information of the black region in the center of the binary image, measuring $100 \times 100$ pixels, is considered the distance between the depth camera and the conveyor belt. By subtracting the height ($h$) from the non-zero elements of the matrix ($BWD$), a corresponding height matrix ($H$) is obtained for each pixel block. Finally, the volume ($V$) of the sand that leaked outside the tray can be calculated using Equation (3).

$$H \odot S = V \tag{3}$$

### 2.7.3. Sand Surface Flatness

To calculate the flatness of the sand, it is necessary to localize the region of the sand. However, localizing this region can be a challenging task, mainly due to the mirror effect caused by the metal tray wall. This effect creates noise that closely resembles the appearance of the sand itself. In order to overcome this challenge, this study employed a combination of edge detection and color thresholding techniques. This approach enabled the segmentation of the sand region located in the middle of the tray with greater accuracy and precision.

First, RGB images (Figure 8a) are segmented using color thresholding. It is visible that the plate and the sand are brighter than other areas. This is because the HSV color space is more closely related to human perception of color than the RGB color space and can effectively distinguish the brightness and saturation levels of colors. Therefore, this study performed segmentation in the HSV color space and generated a mask image. After masking out the non-target areas at the edges accordingly, the plate's and sand's overall region can be obtained. The built-in function in MATLAB® was used to fill the binary image and remove the regions with fewer than 500 pixels (Figure 8c).

The RGB image is first subjected to a Gaussian filter using a $7 \times 5$ convolution filter, and the blurred image is then processed using the Canny operator for edge detection. Two different thresholds were set to detect edges in the image. The first threshold (set to 5) detects obvious edges, while the second threshold (set to 50) connects these edges. The convolution kernel is iteratively selected by trial and error to achieve the best results. This operation is performed three times using a $9 \times 9$ convolution kernel, each iteration using the same kernel size. After processing, a final binary image, the edge map, is obtained (Figure 8d). However, edge discontinuity is a common phenomenon in edge detection algorithms, and the Canny algorithm does not perform well in noise. After obtaining the edge map, this experiment also performed dilation and erosion operations. The dilation operation mainly increases the area of pixels with a value of 1 (white pixels). In contrast, the erosion operation mainly increases the size of pixels with a value of 0 (black pixels).

- By observing the image of the results, it can be seen that the unconnected parts of the edges are mainly concentrated in the lower part and the four corners. Therefore, it requires appropriately modifying the convolution kernel of the dilation operation to link the edges. After testing, a convolution kernel with a size of $9 \times 7$ was selected. This convolution kernel can dilate more in the horizontal direction and less in the

vertical direction while maintaining connectedness among all horizontal lower edges and achieving the highest possible boundary accuracy.

- To ensure the edge is as precise as possible, it is necessary to perform an erosion procedure after the dilation operation to reduce its expansion. After extensive testing, a convolution kernel with a size of $7 \times 6$ was eventually chosen for the erosion procedure. The reason for selecting an asymmetric convolution kernel with unequal sides is to correspondingly reduce the width of the edge caused by the dilation and to follow the dilation convolution kernel after trial and error (Figure 8e). Next, the binary image of the overall region of the plate and sand mentioned above is added using matrix addition. The image segmentation detects areas where pixel values are not equal to 1 and sets them to 0. The corresponding area of sand in the tray can be obtained by directly finding the largest connected region in the extracted image (Figure 8f). Then, denoising and edge smoothing operations are performed to obtain the image with the mask, as shown in (Figure 8g).
- In the last step, the sand region's binary image can be obtained. Then, the corresponding depth information of the sand in the tray can be obtained by taking the dot product of the binary image and depth point cloud information (Equation (4)). After denoising the point cloud information, the standard deviation is calculated to determine the flatness of the sand in the tray from the corresponding image.

$$I \odot D = SD \tag{4}$$

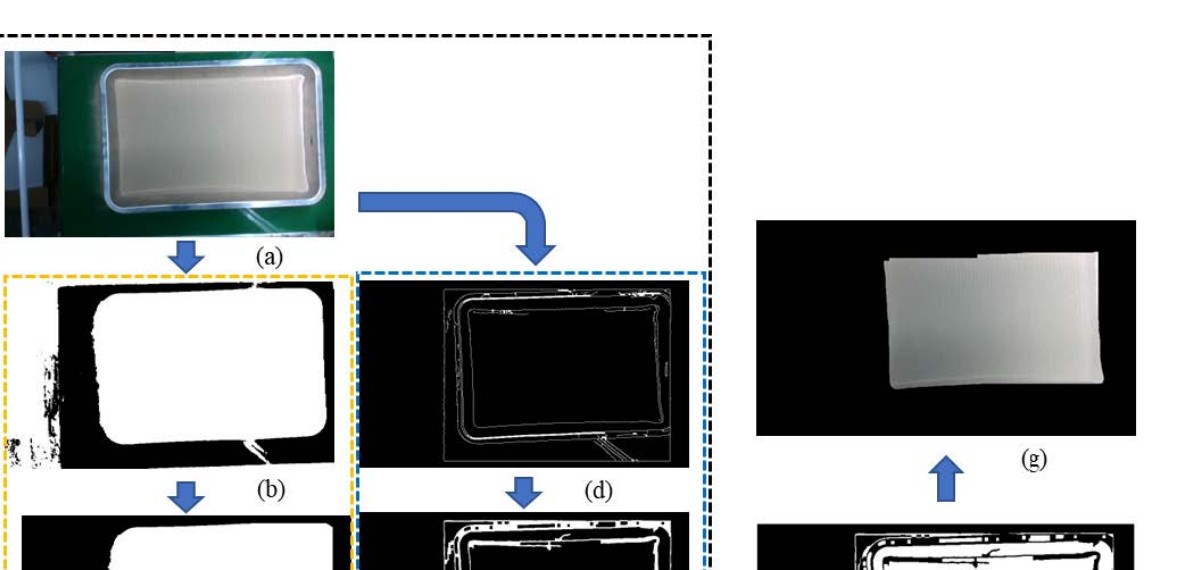

**Figure 8.** Flatness process: (**a**) RGB image; (**b**) Image after color threshold segmentation; (**c**) Image with masked edges; (**d**) Edge detection; (**e**) Dilation and erosion; (**f**) Image obtained by performing bitwise OR operation on two images; and (**g**) Target region.

### 2.8. Determining the Satisfactory Factor Combination

This experiment used the analytic hierarchy process (AHP) to discover a good combination of the three factors (i.e., sand weight in the tray, sand volume left on the conveyor, and sand surface flatness). AHP is a technique that may prioritize decisions based on pairwise comparisons and expert assessment [22]. The primary process of using AHP

is as follows: (1) data normalization; (2) establishing a judgment matrix; (3) consistency analysis; (4) calculating weights; and (5) calculating scores for each factor level combination. Normalization must be conducted as the first step to account for the significant difference in data magnitude between the three evaluation indicators. However, as weight serves as a positive indicator (where higher values indicate better performance), it is necessary to perform positive indicator normalization using Equation (5). Conversely, as volume and standard deviation act as negative indicators (where lower values indicate better performance), a negative indicator normalization using Equation (6) is required. Finally, the normalized values of the three indicators (weight, volume, and flatness) are obtained.

$$X_{normal} = \frac{X - X_{\min}}{X_{\max} - X_{\min}} \tag{5}$$

$$X_{normal} = \frac{X_{\max} - X}{X_{\max} - X_{\min}} \tag{6}$$

A judgment matrix was created using Saaty's scoring method to compare each pair of indicators, and the corresponding scoring criteria are presented in Table 1. Saaty's scoring criteria offer a clear and accurate evaluation of the level of different factors, and they are widely adopted because of their ease of comprehension. Based on the scores given by three experts and taking the median value, it was determined that the importance of flatness to weight and volume are 7 and 4, respectively. On a scale of 1 to 9, the importance of volume to weight is rated at 3, while flatness is rated at 1/4. Also, the importance of weight to volume and flatness are rated 1/3 and 1/7, respectively. The judgment matrix can be constructed as follows.

$$A = \begin{bmatrix} 1 & 0.33 & 0.14 \\ 3 & 1 & 0.25 \\ 7 & 4 & 1 \end{bmatrix} \tag{7}$$

**Table 1.** Saaty's scoring scale and its explanation.

| Score | Explanation |
|---|---|
| 1 | Equal importance |
| 3 | Moderate importance |
| 5 | Very strong importance |
| 7 | Intermediate between extreme and very strong importance |
| 9 | Intermediate between very strong and strong importance |
| 2, 4, 6, 8 | The importance is somewhere in between |

The maximum eigenvalue of the judgment matrix is $\lambda_{\max} = 3.032$. $RI$ can be found to be 0.58, provided by Saaty's AHP method. Using Equation (8), the consistency index ($CI$) can be calculated using Equation (9), and the consistency ratio ($CR$) can be calculated.

$$CI = \frac{\lambda_{\max} - n}{n - 1} \tag{8}$$

$$CR = \frac{CI}{RI} \tag{9}$$

$$CR = 0.0311 < 0.1 \tag{10}$$

If $CR$ is less than 0.1, it is considered to have passed the consistency check (Equation (10)) in this experiment. Then, the weights can be calculated using the arithmetic mean method, and the weights are shown in Table 2.

**Table 2.** Indicators and their corresponding weights.

| Indicators | Weights |
| --- | --- |
| Weight | 0.086 |
| Volume | 0.213 |
| Flatness | 0.701 |

By utilizing the weights derived from the AHP method, Equation (11) can be employed to compute the scores for all possible combinations of conditions.

$$score = 0.086 \times W + 0.213 \times V + 0.701 \times F \tag{11}$$

The experiment repeated the same factor level combination three times to prevent data randomness. The average values of weight, volume, and flatness indicators from three repetitions were used for those indicators in that factor level combination. Table 3 shows the experimental factors and levels.

**Table 3.** Factors and levels of the experiment.

| Level | Factor A: Gate Opening Size (mm) | Factor B: Sensor Location (mm) | Factor C: Conveyor Speed (mm/s) |
| --- | --- | --- | --- |
| 1 | 7.2 | 40.0 | 13.0 |
| 2 | 9.0 | 70.0 | 51.0 |
| 3 | 10.8 | 90.0 | 65.0 |

## 3. Results and Discussion

### 3.1. Weight of Sand in the Tray

Figure 9 shows the weight of sand in the tray under different factor combinations. It can be seen that when Factor A (gate opening size) and Factor C (conveyor speed) remain constant, the change in Factor B does not result in a significant weight variation, which means that the distance of the sensor has little effect on the weight of the sand in the tray. We calculate the standard deviation of Factor B under the same conditions as Factor A and Factor C. The standard deviations, relative to the mean of Factor B, are generally lower than 15%. The size of the opening plays a crucial role, particularly when it reaches the maximum opening size (A3). At this point, there is a significant difference in the weight of sand present in the tray compared to the two other opening sizes (A1 and A2). Additionally, the speed of the conveyor belt has a notable effect on the sand in the tray. As the conveyor belt speed increases, the weight of the sand in the tray decreases.

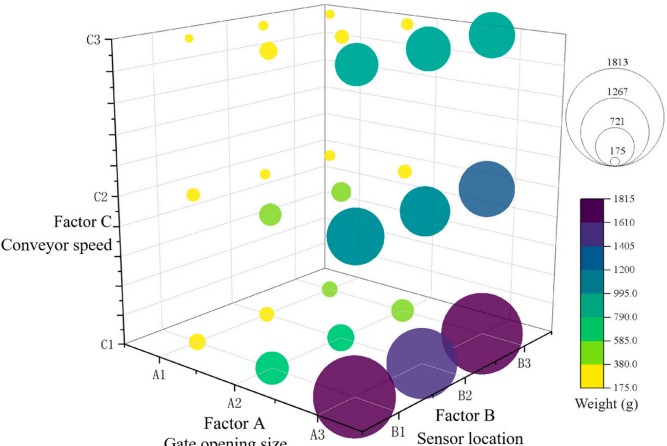

**Figure 9.** Weight data of sand in experimental trays for each group. (Factor A: Gate opening size; Factor B: Sensor location; and Factor C: Conveyor speed).

*3.2. The Volume of Sand Outside the Tray*

Figure 10 shows the volume of sand under different factor combinations. The volume of sand outside the tray (left on the conveyor) increases when the conveyor belt speed increases. When the conveyor belt speed is fast, the tray moves quickly and may exit the gate area before the gate is fully closed, causing sand to be left on the belt. Furthermore, the positioning of the sensor is also a significant factor affecting the performance. Placing the sensor closer to the device greatly reduces the volume of sand leaked outside the tray, while placing it farther away results in more sand leakage. When the gate opened before trays are delivered under the device, a large amount of sand will fall on the conveyor. If the device aperture is tiny and the sensor is located near the device, a large amount of sand will be wasted. This can occur even when the conveyor belt speed remains constant. On the other hand, the amount of sand lost is less if the device hole is wide and the sensor is positioned further away.

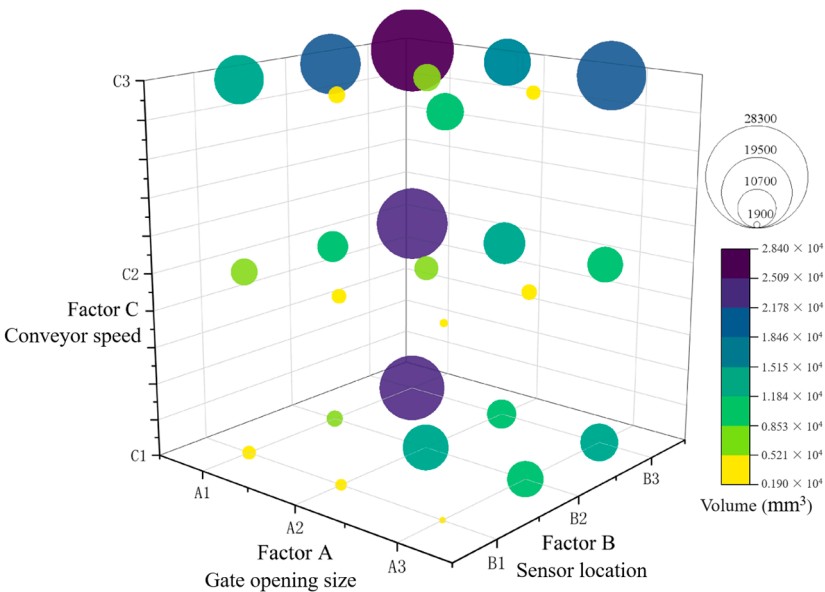

**Figure 10.** Volume data of sand leaked outside the tray for each group of experiments. (Factor A: Opening Size; Factor B: Sensor Location; and Factor C: Conveyor Speed).

*3.3. Flatness of Sand Surface in the Tray*

Figure 11 illustrates the variation in flatness across different factor combinations. The flatness of the sand within the tray is observed to be poor in cases where the opening size is either the smallest or the largest. This may be attributed to the uneven distribution of sand when the opening is small, and a higher quantity of sand falling simultaneously when the opening is wide. Moreover, the contact force between sand particles and the subtle vibrations resulting from the motor movement can also influence the final flatness of the sand within the tray. The experimental data indicate that, on average, the flatness level is better when using a moderate opening size (A2) compared to the other two sizes. Additionally, it is observed that a slower conveyor belt speed can yield a relatively flat sand surface when the opening is small. Conversely, a faster conveyor belt speed is required when dealing with a larger opening size to achieve a relatively flat sand surface in the tray. When the sensor is placed closer, the flatness of the sand in the tray tends to decrease. Conversely, when the sensor is positioned farther away, the flatness improves. This occurs because when the sensor is in close proximity to the sand outlet, the tray enters beneath the outlet before the device reaches the desired size, resulting in uneven sand distribution in front of the tray.

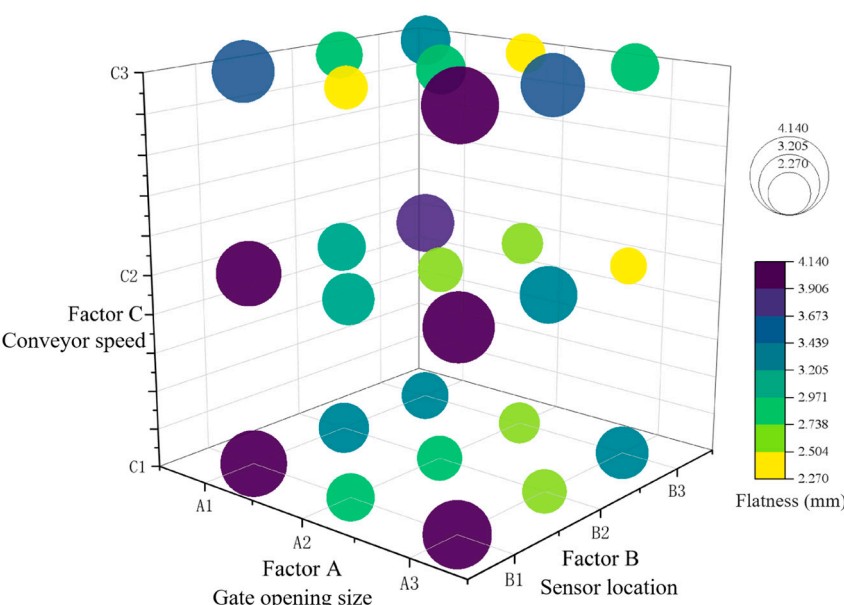

**Figure 11.** Data of sand flatness for each group of experiments (Factor A: Opening Size; Factor B: Sensor Location; and Factor C: Conveyor Speed).

*3.4. Satisfactory Combination*

Two combinations, A3B3C2 and A2B1C3, both yield scores above 0.8 when calculated using Equation (11). The combination A3B3C2 corresponds to an opening size of 10.8 mm, a sensor distance of 9.0 cm, and a conveyor belt speed of 5.1 cm/s. Similarly, the combination A2B1C3 represents an opening size of 9.0 mm, a sensor distance of 4 cm, and a conveyor belt speed of 6.5 cm/s. Various score values were tested based on the expert-assigned importance rules (where flatness is deemed more important than volume, and volume is considered more critical than weight). However, even with different score values, the combinations A3B3C2 and A2B1C3 consistently achieved the highest scores. According to our scoring calculation method, this experiment identifies A3B3C2 as the best combination while acknowledging that A2B1C3 is also a favorable choice. During the experiment, it was noticed that the flatness of the sand surface is affected by the sand-filling rate and the speed at which the tray moves (conveyor speed). The sand-filling rate is controlled by the gate opening size. Thus, regarding the sand surface flatness, the gate opening size and the speed of the conveyor are the two factors requiring investigation. Sand that falls onto the belt, not in the tray, is wasted. Adjusting the timing of the device's opening and closing can help reduce the amount of wasted sand. Since the time adjustment is related to the location of the sensor, the sensor installation place is another factor requiring investigation. Thus, the effects of three factors (i.e., gate opening size, conveyor speed, and sensor mounting location) on the machine performance should be studied.

**4. Conclusions**

This study focused on the development of an automated seeding machine, which involved four distinct phases: sand paving, seed positioning, watering, and sand covering. The performance of the machine was influenced by three factors: conveyor belt speed, gate opening size, and sensor horizontal distance from the sand gate. To evaluate its performance, three parameters were considered: the weight of sand in the tray, the volume of sand outside the tray, and the flatness of the sand. The analytic hierarchy process was employed to determine the optimal combination by exploring the relationship between these three assessment indicators.

The evaluation process placed significant emphasis on flatness due to its negative impact on the early germination rate detection of maize seeds. Considering the experiment's outcomes and utilizing the AHP approach, the following combination of factors is recom-

mended for achieving the most satisfactory performance: a gate opening size of 10.8 mm, a horizontal sensor distance of 9 cm from the gate, and a conveyor speed of 5.1 cm/s. This particular combination proves to be optimal for the automated seeding machine, ensuring the uniform distribution of a substantial volume of sand and maintaining desirable flatness.

The sand-spreading device examined in this experiment is a self-contained module with the potential for application in various related fields. This cost-effective and straight-forward device offers the possibility of reducing expenses and enhancing profitability for companies. As a result, it holds promise for diverse industries that necessitate accurate and uniform sand or granular material distribution. Furthermore, the methods and evaluation apparatus employed in this study can be extrapolated to larger, more intricate sand-spreading machinery, offering valuable reference values for advancing precision agriculture's development.

**Author Contributions:** Conceptualization, B.F. and Z.Z.; methodology, Z.Z.; software, B.F.; validation, B.F., W.S. and Z.Z.; formal analysis, B.F.; investigation, B.F.; resources, Z.Z.; data curation, B.F.; writing—original draft preparation, B.F.; writing—review and editing, W.S. and Z.Z.; visualization, B.F.; supervision, Z.Z.; project administration, Z.Z.; funding acquisition, Z.Z. All authors have read and agreed to the published version of the manuscript.

**Funding:** This research is supported by the Chinese Universities Scientific Fund (funding no. 15053343).

**Data Availability Statement:** The data presented in this study are available on request from the corresponding author.

**Conflicts of Interest:** The authors declare no conflict of interest.

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
