# Peer review of "Optimal Sand−Paving Parameters Determination of an Innovatively Developed Automatic Maize Seeding Machine"

_agriculture, doi:10.3390/agriculture13081538_

Round 1
Reviewer 1 Report
This study developed an automatic seeding system and focused on determining the most satisfactory factor combinations for paving sand. Data is obtained by using an RGB-D camera, and image processing technology is used to obtain the amount of retained sand and the sand surface smoothness. Three-variable and three-level full factorial experiments were designed to determine the most satisfactory factor combination. This research can provide some guiding information for the practical application of automatic seeding machine. Overall, this paper is a comprehensive study through well-designed experiments. The findings are clear and concise. However, for some problems, improvements are still needed.
Specific comments on the paper:
1. Line 17-18: In the sentence "Our group developed an automatic maize seeding machine", our group is not an appropriate expression, please use "this research" or other more appropriate expressions. 2. Line 19-20: In the sentence "performance is significantly affected", consider rewriting it as "performance is affected". 3. Line 26: What system does "The system" refer to? Please explain in details. 4. Line 39-43: Please streamline and reorganize the presentation. 5. Line 39-43: "Our team" in the text, it is suggested to change to a more appropriate Tibetan way, such as the design of this study. 6. Line 111: In the article, "29.7cm x 29.0cm x 40.0cm," is recommended to be unified as "mm". 7. Line 113: The format of "24 VDC" is wrong, it is recommended to modify. 8. Line 115-120: Please reorganize the language. 9. Line 132: The speed unit is recommended to use the international standard. 10. Line 132: Please describe in detail the data types processed by MATLAB, such as image data, etc. 11. Line 155-160: Using this method, does the calibration effect receive the effect of ambient light? 12. Line 179-182: Why is area calculated here? Consider deleting if subsequent methods are not utilized. 13. Line 184: "510mm", it is recommended to use m as the unit. 14. Line 207-208: Image format problem, it is recommended to center the image. 15. Line 380-381: Please pay attention to the typesetting of the paper, it is recommended not to have a large blank space. 16. Line 429:“This study developed an automated seeding system”,The abstract research is automatic seeding machinery, the conclusion is the research of automatic seeding system, the author is requested to clarify the research contentAuthor Response
Please see the attachment.

Reviewer 2 Report
The paper:
Optimal Sand Paving Parameters Determination of an Innovatively Developed Automatic Maize Seeding Machine
Was reviewed giving the following results:
Eliminate are before the word landed in line 52.
End line 55 after sand surface flat.
In line 56 add the after during and before experiment. All the paragraph (lines 56-65) should be moved to line 427 before the conclusion section.
Lines 89 and 90 are the same as 91 and 92 and one has to be removed.
Change in line 81 the word and by the word to in “evaluation and deter”
Please rewrite lines 104-106 as the meaning is not clear.
Lines 115 to 117 are not well explained and should be rewritten.
Change in line 151 “most concerned” by more important.
Change in line 195 noises for noise.
In figure 10, it will be clearer to post in each axis the variable, weight, conveyor speed and sensor location.
Change with by when in line 386.
Change in figure 11 and 12 the axis names.
THE ARTICLE IS EXCELENT BUT ALTOUGH YOU MENTION 4 PHASES DONE BY THE MACHINE YOU ONLY TEST SAND PAVING. WHAT HAPPENS DURING WATERING; PLANTING; ETC. WAS IT EXACT?
References
In reference 3 the word proceedings is repeated twice, and the place of the symposium was not stated.
Reference 11 also lacks place of symposium.
Reference 16 and 21 is a book and it hasn’t the editorial and country.
There are some grammar issues which were mentioned in the previous file.
